# NHS CHECK: protocol for a cohort study investigating the psychosocial impact of the COVID-19 pandemic on healthcare workers

Danielle Lamb ,[1] Neil Greenberg,[2] Matthew Hotopf,[2] Rosalind Raine,[1] Reza Razavi,[2] Rupa Bhundia,[2] Hannah Scott,[2] Ewan Carr ,[2] Rafael Gafoor,[1] Ioannis Bakolis,[2] Siobhan Hegarty,[2] Emilia Souliou,[2] Anne Marie Rafferty,[2] Rebecca Rhead,[2] Danny Weston,[2] Sam Gnanapragasam,[2] Sally Marlow,[2] Simon Wessely ,[2] Sharon Stevelink[2]

[1]Department of Applied Health Research, University College London, London, UK
[2]Institute of Psychiatry, Psychology & Neuroscience, King's College London, London, UK

**Correspondence to**
Dr Danielle Lamb;
d.lamb@ucl.ac.uk

## ABSTRACT

**Introduction** The COVID-19 pandemic has had profound effects on the working lives of healthcare workers (HCWs), but the extent to which their well-being and mental health have been affected remains unclear. This longitudinal cohort study aims to recruit a cohort of National Health Service (NHS) HCWs, conducting surveys at regular intervals to provide evidence about the prevalence of symptoms of mental disorders, and investigate associated factors such as occupational contexts and support interventions available.

**Methods and analysis** All staff, students and volunteers working in the 18 participating NHS Trusts in England will be sent emails inviting them to complete a survey at baseline, with email invitations for the follow-up surveys sent 6 months and 12 months later. Opening in late April 2020, the baseline survey collects data on demographics, occupational/organisational factors, experiences of COVID-19, validated measures of symptoms of poor mental health (eg, depression, anxiety, post-traumatic stress disorder), and constructs such as resilience and moral injury. These surveys will be complemented by in-depth psychiatric interviews with a sample of HCWs. Qualitative interviews will also be conducted, to gain deeper understanding of the support programmes used or desired by staff, and facilitators and barriers to accessing such programmes.

**Ethics and dissemination** Ethical approval for the study was granted by the Health Research Authority (reference: 20/HRA/210, IRAS: 282686) and local Trust Research and Development approval. Cohort data are collected via Qualtrics online survey software, pseudonymised and held on secure university servers. Participants are aware that they can withdraw from the study at any time, and there is signposting to support services if participants feel they need it. Only those consenting to be contacted about further research will be invited to participate in further components. Findings will be rapidly shared with NHS Trusts, and via academic publications in due course.

## Strengths and limitations of this study

► The longitudinal cohort design addresses the lack of long-term data on this population, and the current predominance of cross-sectional evidence available.
► The availability of Trust Human Resources data means we will be able to calculate response rates, and weight the data appropriately.
► The diagnostic interview component of the study will allow us to establish the true prevalence of mental disorders, which can be inflated by the measures used in most mental health and well-being cohort studies.
► The qualitative interviews will give deeper insight into the support programmes that healthcare workers (HCWs) find most helpful, and provide ideas for Trusts to improve their offer to staff.
► The three components of the study give breadth and depth lacking in much of the mental health and well-being research currently available, but there is a risk of overburdening already stretched HCWs.

## INTRODUCTION

The COVID-19 pandemic raises many questions on biological, behavioural, emotional and social responses to a global threat.[1] Evidence from systematic reviews and meta-analyses suggests that healthcare workers (HCWs) who have had to deal with serious infection during pandemics in the past (severe acute respiratory syndrome (SARS), Middle East respiratory syndrome (MERS), Ebola, swine influenza) are at increased risk of both current and subsequent mental health problems.[2 3]

Pandemics expose HCWs to overwork, isolation from friends and family, discrimination, exhaustion and an increased risk of developing common mental disorders.[3] Early

BMJ

accounts from Wuhan, China, confirm this, suggesting that the added pressures of providing healthcare during a pandemic can result in impaired decision making, attention and understanding thereby hindering the control of the pandemic, but also early signs of distress may well lead to longer-term mental ill-health.[4] A relatively new concept that has attracted a lot of attention is the concept of 'moral injury'.[5] Moral injury describes the psychological distress resulting from actions, or the lack of them, which violate someone's moral or ethical code and can contribute to the development of mental health difficulties, including depression, post-traumatic stress disorder (PTSD) and suicidal ideation.[6] During the COVID-19 pandemic many National Health Service (NHS) staff may have had to make difficult choices not faced before, to deliver care that they know is suboptimal and to explain difficult decisions to relatives. Such ethical dilemmas will be new in scale and nature.

Research to date suggests that women, younger people, and those from racial and ethnic minority groups are at higher risk of adverse outcomes.[7] There is evidence that those in lower income brackets have been more negatively affected,[8] and that nurses may be worse affected than those in other roles.[9] In terms of trajectory, well-being appears to have worsened during the early stages of the pandemic, with small improvements through the following months.[8] Other factors associated with poor mental health and well-being are lack of access to personal protective equipment (PPE) and lack of supportive environments.[10] While research on moral injury has previously focused on military contexts, early work in healthcare settings shows concerning associations with poorer outcomes.[10 11]

Much of the existing research in this area is based on single questionnaire assessments, typically only including clinical staff rather than all HCWs, from which bold claims of mental health crisis are made. While survey data can be informative, two-stage epidemiological studies using diagnostic interviews tend to show that such surveys overestimate prevalence of mental disorders.[12] To address these issues, our study has five distinct features: (1) Breadth geographically, with a range of different types of NHS Trusts (eg, acute, mental health) and locations (urban, rural, all areas of England); (2) Depth phenotypically, with diagnostic interviews included in order to ascertain true prevalence of mental disorders rather than simply indicators of distress; (3) Longitudinal data collection to look at temporal patterns, meaning we will be able to identify whether a surge of symptoms at time of crisis lead to persistence or remission; (4) Inclusivity regardless of role, meaning that we will include all HCWs who are contributing to the pandemic effort, whether they are in clinical or ancillary and support roles; and (5) The ability to calculate accurate response rates and weight data appropriately, thanks to demographic population data provided by each participating Trust's Human Resources department.

## AIMS, OBJECTIVES AND HYPOTHESES

The main aim of the study is to investigate the psychosocial impacts of the COVID-19 pandemic on NHS Trust workforce mental health and well-being over time. In addition, we aim to explore the uptake and usefulness of staff support interventions available to participants.

The primary objective is to establish a cohort of all staff employed in participating NHS Trusts, to carry out repeated surveys of their mental health and well-being and psychiatric diagnostic interviews to determine the true prevalence of disorders. Based on the literature outlined above, we have a number of descriptive aims and hypotheses. We will:

1. Describe the prevalence of psychological distress and characteristics associated with poorer mental health. Hypotheses include:
   - *Poorer mental health will be associated with demographic variables (including younger age, female sex, coming from Black, Asian, or other racial or ethnic minority groups).*
   - *Poorer mental health will be associated with occupational characteristics (including role, pay grade, work setting, redeployment status).*
2. Establish the true prevalence of mental disorders via diagnostic interviews.
   Secondary objectives include exploring the factors associated with poor mental health, patterns of reported distress over time, and qualitatively exploring experiences of staff support interventions. We will:
3. Describe workplace factors associated with poorer mental health. Hypotheses include:
   - *Poorer mental health will be associated with higher levels of reported moral injury.*
   - *Poorer mental health will be associated with reported lack of access to PPE.*
   - *Poorer mental health will be associated with perceived lack of support from leaders, team, friends/family.*
4. Describe patterns and persistence of psychological distress symptoms over time. Hypotheses include:
   - *Mental health and well-being will follow the trajectory of the pandemic, with poorer outcomes evident during/after higher levels of COVID-19 prevalence (eg, as measured by daily deaths, hospital admissions).*
   - *Poorer mental health at baseline will be associated with poorer mental health at 6-month and 12-month follow-up points.*
   - *Predictors of poorer mental health at follow-up time points will include younger age, female sex, racial or ethnic minority group, being a nurse, perceived lack of support, lack of access to PPE, and higher levels of reported moral injury.*
5. Qualitatively evaluate tiered and tailored staff support programmes being implemented locally and nationally, using in-depth interviews and thematic analysis. This will enhance our understanding of how we could further scale up effective support programmes within and across Trusts.
6. Provide a platform for further randomised controlled trials (RCTs), and observational or intervention studies, including:

- An ethnic inequalities module, which will explore ethnic inequalities in mental health and occupational outcomes across NHS staff and the mechanisms that perpetuate these inequalities, using a mixture of quantitative and qualitative data (Tackling Inequalities and Discrimination Experiences in health Services Study: TIDES https://tidesstudy.com).
- Estimation of parameters of interest which could be later incorporated when designing an RCT or a pragmatic trial, for example, prevalence of psychosocial distress, intraclass correlation coefficient across the NHS Trust.
- An RCT testing a well-being app for use by NHS staff.

## METHODS
### Design and setting
This study consists of three main components:
1. A longitudinal cohort study, consisting of surveys administered at baseline, 6 months and 12 months, to track outcomes over time.
2. A diagnostic interview study, using a clinical diagnostic measure to ascertain the true prevalence of mental disorders.
3. A qualitative interview study, using semistructured interviews to explore experiences of using (or reasons for not using) staff support programmes.

The study will be carried out in 18 NHS Trusts in locations across England.

### Study population and sample
#### Longitudinal cohort study
Participants across the study will be any NHS-affiliated staff including clinical staff, students and all support staff working within participating NHS Trusts: Avon and Wiltshire Mental Health NHS Foundation Trust (n=4334); Cambridge University Hospitals NHS Foundation Trust (n=10 243); Cambridgeshire and Peterborough NHS Foundation Trust (n=4235); Cornwall Partnership NHS Foundation Trust (n=3977); Devon Partnership NHS Foundation Trust (n=3280); East Suffolk and North Essex NHS Foundation Trust (n=10 219); Gloucestershire Hospitals NHS Foundation Trust (n=8437); Guys and St Thomas' NHS Foundation Trust (n=19 760); King's College Hospital and Princess Royal University Hospital (PRUH) (n=12 959); Lancashire and South Cumbria NHS Foundation Trust (n=6984); Norfolk and Norwich University Hospitals (n=10 502); Nottinghamshire Healthcare NHS Foundation Trust (n=8860); Royal Papworth Hospital (n=2110); Sheffield Health and Social Care (n=2610); South London and Maudsley NHS Foundation Trust (n=5151); Tees Esk and Wear Valleys NHS Foundation Trust (n=7315); University Hospitals of Derby and Burton (n=13 231); University Hospitals of Leicester NHS Foundation Trust (n=16 946); and/or any of the Nightingale Hospitals (London, Exeter, Leeds/Harrogate, Cardiff and Manchester), dependent on whether these sites are active at the time of data collection. Study sites were selected to cover multiple areas of England. All staff in each study site will be invited to participate.

### Diagnostic interview study
Participants will be up to 350 HCWs who have completed the baseline NHS CHECK survey.

Up to 250 participants will be purposively sampled according to their responses to the General Health Questionnaire (GHQ) in the short survey module;[13] half will meet caseness and half will not meet caseness. These participants will be administered the Clinical Interview Schedule-Revised (CIS-R).[14] The number of CIS-R interviews (n=250) has been chosen balancing precision and cost. Based on a simulation study, we anticipate that 250 interviews will allow estimation of CIS-R prevalence of common mental disorders to within ±6%.

Up to 100 HCWs will be sampled according to their responses to the 6-item PTSD Checklist civilian version (PCL-6)[15] in the long survey module; half will meet caseness and half will not meet caseness. These participants will be administered the Clinician-administered PTSD Scale (CAPS-5).[16] This sample size was calculated as the necessary number to achieve a valid estimation of PTSD prevalence.

Sampling for the CIS-R and CAPS-5 groups will be stratified to ensure representation from each of the 18 participating sites in the NHS CHECK Study, as well as to ensure ethnic, sex and age breakdown that resembles respondents of the baseline survey.

### Qualitative interview study
Participants will be sampled from two groups: (1) Up to 48 participants who complete the NHS CHECK 6-month follow-up survey; and (2) Up to 12 members of staff (from participating Trusts) who are involved in implementing staff support programmes, who do not need to have completed any NHS CHECK surveys. Group 1 will be sampled according to sex, ethnicity, age (a cut-off of 50 years will be used to dichotomise participants into younger or older groups), and occupational role. Balancing these demographic factors, four discrete groups of HCWs will be sampled, with 12 in each group: (i) Those who used support programmes and found them helpful; (ii) Those who used support programmes but did not find them helpful; (iii) Those who heard about support programmes but did not use them; and (iv) Those who had not heard about support programmes. We will endeavour to include a diverse range of participants in Group 2, but this will be dependent on those involved in staff support in participating Trusts.

Participants in Group 1 will be recruited first, and what they tell us will inform subsequent interviews with those in Group 2.

### Inclusion criteria
Inclusion criteria for each part of the study are as follows.

### Longitudinal cohort study

Participant must:

1. Be an NHS-affiliated member of staff, working at, or with (eg, volunteer or student), the participating NHS Trusts and/or Nightingale Hospitals during the COVID-19 pandemic.
2. Be aged 18 and over.
3. Be able to give informed consent to take part in research.
4. Be able to understand and communicate in English.
5. Have access to an email address to facilitate survey registration and receive follow-up survey links.

### Diagnostic interview study

Participants must meet the criteria for the cohort study, and must also:

1. Have completed the baseline NHS CHECK Survey as an NHS member of staff based at a participating Trust; including the relevant PTSD survey measures if being administered the CAPS-5.
2. Have indicated in the baseline survey that they consent to be contacted for participation in further research.
3. Have access to a phone for the interview.
4. Have scored ≥4 on the GHQ to meet caseness for probable common mental disorders or ≥14 on the PCL-6 to meet caseness for probable PTSD.

### Qualitative interview study

Participants in Group 1 must meet the criteria for the cohort study, and must also have:

1. Completed the baseline and 6-month follow-up surveys of the longitudinal cohort study.
2. Indicated in the baseline survey that they consent to be contacted for participation in further research.
3. Provided data on whether or not they have heard of and/or used any staff support programmes.

Participants in Group 2 must:

1. Have been involved in implementing staff support programmes in one of the 18 participating Trusts or Nightingale hospitals.
2. Be aged 18 years and over.
3. Be able to give informed consent to take part in research.
4. Be able to understand and communicate in English.

### Measures

#### Longitudinal cohort study

##### Baseline

The baseline survey will involve a short survey (5–10 min) which collects information on the following topics: (1) Contact details, (2) Occupational information (eg, occupational group, length of professional registration), (3) Sociodemographic characteristics, (4) Working practices (eg, access to PPE, performing aerosolising procedures), (5) Perceived support from managers, colleagues, friends and family, (6) COVID-19-related questions (eg, suspected infection status, COVID-19 test status, isolation/quarantining), (8) Staff support programme access,

(9) Self-reported diagnosed health conditions and (10) Common mental disorders. The prevalence of probable common mental disorders will be assessed using the 12-item GHQ-12, with a cut-off score of 4 or more indicating caseness.[13]

There will be the option for participants to complete an additional longer survey (10–15 min), which includes information on the following: (1) Impact of COVID-19 (eg, on family, income, health, positive and negative changes in personal life or work), (2) Work experiences (leadership and teamwork, sickness absence, unsafe clinical practices, preparedness), (3) Usefulness of staff support programmes, (4) Caring responsibilities outside of work, (5) Confidence in institutions to handle the COVID-19 pandemic. In addition, the following validated measures will be used in the longer survey: the 7-item Generalised Anxiety Disorder (GAD-7) Scale to measure probable moderate anxiety disorder with a cut-off score of 10 or more indicating caseness;[17] the 9-item Patient Health Questionnaire (PHQ-9) to measure probable moderate depression with a cut-off score of 10 or more indicating caseness;[18] the 10-item Alcohol Use Disorder Identification Test to measure alcohol consumption with a cut-off score of 8 or more indicating hazardous drinking;[19] the PCL-6 civilian version to measure PTSD with a cut-off score of 14 or more indicating the presence of probable PTSD;[15] the 9-item Moral Injury Event Scale (MIES) to measure moral injury, with a higher score indicating greater exposure to morally injurious events;[20] the 14-item Warwick-Edinburgh Mental Well-being Scale (WEMWBS) to measure subjective well-being and psychological functioning, with a higher score indicating a higher level of mental well-being;[21] the 6-item Brief Resilience Scale (BRS) to measure psychological resilience, with a higher score indicating a higher level of resilience;[22] the 12-item Burn-out Assessment Tool (BAT) to measure burn-out, with a cut-off score of 3.02 indicating probable burn-out.[23] All validated measures use Likert Scale response options.

We will also assess suicidal ideation in the longer survey, using items related to suicidal thoughts, suicide attempts and self-harm derived from the CIS-R.[14] We will measure fatigue using three novel items exploring: (1) Emotional and physical exhaustion on a 0–10 scale; (2) Whether any experienced fatigue is greater than usual tiredness (yes/no); and (3) Whether any experienced fatigue interferes with the ability to do things (yes/no).

##### Follow-up—6 months

The 6-month survey will also involve a short version and long version. The short survey will collect information on the national and local staff support programmes used, and on household income. The same questions as at baseline will be asked regarding: COVID-19 experiences (eg, suspected infection status, COVID-19 test status, isolation/quarantining); teamwork; support from colleagues and friends/family. The same measures will be used in the 6-month survey as at baseline, as follows: GHQ-12,

GAD-7, PHQ-9, Alcohol Use Disorders Identification Test (AUDIT), PCL-6, MIES and the CIS-R suicidality questions. In addition, the short form of the Post-traumatic Growth Inventory will be used.[24]

The long survey will collect the same measures as at baseline: WEMWBS, BRS, BAT, Fatigue. It will also collect information on the personal and occupational factors as at baseline. The topics covered may be updated as the pandemic evolves, in order to capture the most relevant information.

### Follow-up—12 months
Similar short and long versions will be used in the 12-month follow-up survey as the 6-month survey, with further refinement as the pandemic evolves.

### Diagnostic interview study
CIS-R will be used to assess mental disorders.[14] The CIS-R is a standardised interview for use in general practice, community settings, hospitals and occupational contexts, and consists of 10 items, each scored on a scale of 0–4, with scores combined to provide a total weighted score. Questions refer to symptoms in the previous week. Use of the CIS-R allows comparison with a national probability sampled survey, the Adult Psychiatric Morbidity Survey.[25]

CAPS-5 will be used to assess PTSD.[16] CAPS-5 is a structured interview for use in clinical and research settings that assesses symptom severity and diagnostic status of PTSD in line with Diagnostic and Statistical Manual of Mental Disorders, Fifth Edition (DSM-5) criteria. The tool consists of 30 items, on a 5-point rating system, across seven criteria that ask about symptoms within the past month; scores are combined to create a total symptom severity score and are used to establish presence or absence of the diagnosis.

### Qualitative interview study
Semistructured interview schedules will be constructed using Normalisation Process Theory as a framework in order to address topics relevant to the evaluation and implementation of interventions.[26] Questions will be included to draw out details about: how and why staff support programmes have been used, including ease of access and use; whether they have been helpful or unhelpful, and why; how information about such interventions could be communicated to staff most helpfully; and whether there are additional supports that would be helpful for Trusts to provide.

### Study procedures
#### Longitudinal cohort study
*Recruitment*
Potential participants will be identified via each Trust Human Resources system. Senior management at each site will use existing dedicated group email lists to circulate details of the study and the URL for the baseline survey of the longitudinal cohort study, emphasising the voluntary nature of participation. A 'cascade' of emails and contacts about the study will be encouraged in each

Trust, via staff support teams/leads, chief nursing officers, medical directors, occupational health departments, Union representatives and well-being hub users. The study will be promoted during team briefings, included in Trust newsletters, highlighted by news items on Trust intranet websites, posted to closed social media groups, set by Information Technology (IT) services as screen-savers on Trust computers, and posters about the study will be displayed in staff rest areas and flyers added in 'goodie' bags for staff (in line with any necessary infection control guidelines). Participants will also be recruited via Trust influenza clinics.

In Nightingale Hospitals the study will be promoted, and its importance highlighted, by the staff health and well-being team leads at each site, as well as the chief nursing officer, medical director and other team leads involved.

### Consent
Potential participants will be able to view the information sheet via the study website, then register and provide informed consent if they wish to complete the survey. Paper copies of the participant information sheet (PIS) and consent form can be provided on request. Questions can be emailed to the research team via a dedicated study email address. Participants will provide consent to take part in the study, including baseline and follow-up surveys, and can opt in to consent to be contacted about any linked future studies (eg, trials of support interventions). We will make clear that participation is voluntary and that participants can withdraw at any time without detriment. Due to the rapid publication of data summaries, it will not be possible to withdraw data from published work, but we will not use withdrawn data in any future publications.

### Data collection
Baseline data were collected from April 2020 to January 2021. Staff completing the baseline survey will be invited to complete follow-up surveys at 6 months and 12 months after they have completed the baseline survey. All participants will be followed up at each subsequent data collection point, regardless of whether they completed the previous survey, unless they choose to withdraw from the study.

The online survey will be administered using Qualtrics survey software, hosted by King's College London, and downloaded data will be pseudonymised and stored on secure servers at King's College London and University College London.

### Diagnostic interview study
*Recruitment*
Participants meeting the inclusion criteria will be emailed with an invitation to participate in the study, including the PIS. Participants who respond to the email indicating their interest will receive a follow-up email containing a link to a digital calendar where they can select an interview slot suitable to them using an assigned participant

Identification (ID), which will be confirmed by email by the research team. Participants who do not select a time slot within 10 days of the booking emails being sent will be prompted to respond if still interested in participating, and followed up by phone calls or texts to confirm interest or withdrawal.

Participants will be able to email the research team via the dedicated study email address with any queries.

### Consent

Participants will be sent the link to a Qualtrics survey, 1–3 days prior to their interview date, which will contain the online consent form in which they provide the phone number that the research team will call them on for the interview. Provided consent is given, the survey will continue to the administration of the GAD-7, PHQ-9, PCL-6 and GHQ-12.

The morning of the interview, a member of the research team will check completion of the survey; participants who have not started the survey, or who have consented to the study but not completed the measures will be prompted to complete the survey. If necessary, their interview will be rescheduled. Participants who have completed the survey will be send a reminder of their interview time.

### Data collection

Interviews will be conducted over the phone by researchers based in a confidential location, with calls recorded on an encrypted audio recorder in case of a need for data verification. Researchers will manually enter responses to interview questions into a preprogrammed Qualtrics survey form, following the standard interview procedure for the CIS-R or the CAPS-5 depending on which group the participant belongs to.

Once the interview is complete, the interviewer will email the participant a £25 voucher to thank them for their time and log this payment in the contact database.

### Qualitative interview study
### Recruitment

Eligible participants for Group 1 will be contacted via the email addresses and/or phone numbers provided while completing the baseline and 6-month follow-up questionnaires. Participants will be invited by email to share their experiences in a qualitative interview. If recruitment is insufficient using email alone, participants will be contacted by phone (either phone call or text message) to ascertain interest in the interview study. Recruitment will continue until approximately 12 individuals have been sampled in each category of the sampling frame.

Eligible participants for Group 2 will be purposively sampled via existing professional contacts known to the research team. Following initial outreach, snowball sampling will be used until 12 professionals, who were involved in the implementation of staff supports in the NHS, have been recruited.

### Consent

Those expressing an interest in participation will be emailed a PIS and consent form, which they can return via email. Once consent has been obtained, participants will be contacted again to arrange a time suitable to them to be interviewed remotely (via MS Teams/Zoom, or telephone). Interviews are expected to last approximately 30–60 min, will be conducted at a time convenient for the participant and will be recorded with an encrypted Dictaphone. Recordings will later be destroyed following transcription, de-identification and pseudo-anonymisation of interview transcripts.

### Data collection

Participants will initially be asked to briefly explain their role and to describe how it did/did not change since the onset of the COVID-19 pandemic. Participants will then be asked about the following: their perceived personal need of support within the NHS and what type of support is needed; their experiences with existing staff support programmes including why they did/did not use them; perceived barriers to access; perceived utility of staff support interventions during the pandemic and in general; other external forms of supports used; suggestions for how supports could be improved in their timing, targeting and content. Before the end of the interview, participants will be given the opportunity to add anything they feel is important that has not been discussed. Participants will receive a £25 Amazon voucher to thank them for their time.

### Study flow chart

Participants will enter the study and be offered the opportunity to participate in different components as outlined in figure 1.

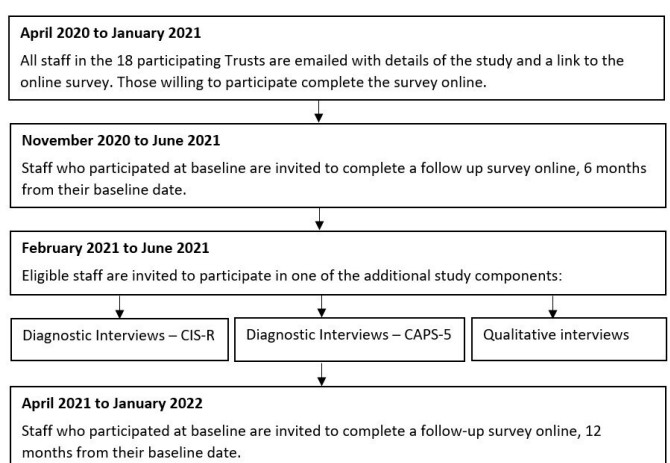

**Figure 1** Flow chart of study timings and components. CAPS-5, Clinician-administered PTSD Scale; CIS-R, Clinical InterviewSchedule-Revised; PTSD, post-traumatic stress disorder.

## Analysis plan
### Longitudinal cohort study
Response weights will be calculated for each Trust, generated using a raking algorithm based on age, sex, ethnicity and role (using Trust population data obtained from Human Resources), with missing demographic data imputed using multiple imputation (for the purposes of weighting only). Trusts with response rates under 5% will be dropped from the analysis. Representativeness of the sample will be described using frequencies and percentages, and descriptive statistics given for each variable to summarise participants (frequencies and weighted percentages for categorical variables, mean and SD for continuous variables). We will examine differences between participants completing only the short survey, compared with those completing both the short and long surveys. We will summarise the weighted prevalence of the primary and secondary outcomes, stratified by sociodemographic and occupational factors. We will explore potential longitudinal associations between sociodemographic characteristics and occupational factors with the outcome measures (eg, GHQ-12, GAD-7, PHQ-9, AUDIT, PCL-6, MIES). Three-level random intercept linear or logistic (depending on the distribution of the outcome) regression models will be used to account for the hierarchical structure of the data, considering observations from baseline, 6-month follow-up and 12-month follow-up at level 2 and NHS trusts at level 3. We will use a measure of pressure on Trusts as the key exposure (eg, ratio of beds in use to staff numbers, ie, those not on sick leave), plotting levels of this exposure over time (aggregated at the Trust level), before partitioning the data into meaningful periods corresponding to burden level over the baseline data collection period (April 2020 to January 2021). We will plot all measures over time to look at patterns between this exposure and outcome levels, and use three-level random intercept logistic and linear regression models, as appropriate, to explore factors associated with outcomes in each period.

### Diagnostic interview study
This study will validate the screening questionnaires used for general distress and PTSDs using diagnostic interviews with the CIS-R and the CAPS-5, respectively. Participants for the validation studies will be randomly selected from those with non-missing screening scores in a 50% ratio of caseness or otherwise. Sample sizes were obtained by simulation studies (n=250 and 100, respectively, for the CIS-R and CAPS-5 diagnostic interviews). Sensitivity, specificity, and positive and negative predictive values will be calculated. If possible, prevalence values for the population at risk will be calculated using population weightings (which will be derived from sex, age and ethnicity variables in an attempt to ensure representativeness in the sample population). Sensitivity analyses will be undertaken to account for missingness if necessary.

### Qualitative interview study
Interviews will be audio-recorded and transcribed verbatim with identifying information removed. Transcripts will be pseudo-anonymised and uploaded to Nvivo V.12 for windows. An inductive qualitative methodology will be used to analyse the interviews underpinned by a pragmatic approach to inquiry. The principles of reflexive thematic analysis will be used to allow an open and organic coding process to develop.[27] Themes will be developed in an iterative process after the initial stages of coding by considering the differences and similarities in the experiences and views of participants from each of the different groups. An initial inductive approach will be applied, more to a more deductive approach over time. A collaborative coding process will be employed, in which members of the research team will initially independently code transcripts and generate a coding framework through discussion. To enhance validity, the emerging thematic framework will be discussed with the wider research team and will be member-checked with several interviewees who consented to be contacted again for this purpose at interview.

### Patient and public involvement
Front-line NHS staff working in intensive care environments proposed this research, having identified a need to rapidly understand and intervene to try and ameliorate the impact of the pandemic on staff. We tested the proposal's acceptability and approach with a small informal reference group of front-line staff (psychologists, managers, intensivists and trainee psychiatrists) and refined it accordingly. We have developed an online advisory group of NHS staff (clinical, managerial, auxiliary, students) and NHS patients to provide input on methods development, recruitment strategy, communications and interpretation of findings. We will also consult this group on tasks such as developing brief lay summaries and interview schedules. We will also run a social media campaign on twitter to raise awareness of the study, as well as to help disseminate our aims, work and results while restrictions on face-to-face events are in place. This will be accompanied by a poster campaign to raise awareness, and increase recruitment to the study, with posters being displayed in sites across all the trusts in the study.

## ETHICAL CONSIDERATIONS
### Longitudinal cohort study
Once consented into the study, an automatically assigned ID number will be used for each participant, allowing survey data to be held pseudonymously. All study staff will adhere to relevant data protection regulations, and will maintain confidentiality at all times. No information that could identify any individual participant will be used in reports or publications, or passed on to the Trusts.

Participants will be able to stop the survey at any point, and can skip questions wherever desired. The only required questions are email address and main employing

Trust. Signposting information will be provided for any participants who experience distress in answering survey questions (eg, links to websites for NHS front-line staff, Mind, Samaritans and the WHO resources for dealing with psychosocial considerations during the pandemic).

There might be some indirect benefits to the participant when taking part. People often value the opportunity to share their opinions, experiences and feelings, and several free-text options throughout the surveys offer the chance to do this.

### Diagnostic interview study

Participants will be asked to discuss their mental health, which can sometimes cause distress. Participants can stop interviews at any point for a break, postpone the interview to another time or day, or end the interview completely. The researcher will provide immediate emotional support, offer to pause or postpone the interview, and offer to contact a friend, family member or other supportive person for participants who show signs of distress. The same signposting information outlined above will be provided to any participants who request this. The research team does not take clinical responsibility for research participants in this study; this is made clear during the consent process. However, a standard risk protocol with a supervising clinician will be followed for participants who indicate that they are at risk of harming themselves or others at any stage of the recruitment or participation process.

As with the survey, there may be some benefits to taking part. People often find it helpful talking about their own experiences. The information gained from the study will be used to inform immediate and future responses to the pandemic, and some people enjoy knowing that they have contributed to this.

### Qualitative interview study

Despite the focus of the interview being on evaluating staff supports, some participants may experience distress in answering questions that draw on their experiences of the COVID-19 pandemic. The nature of the interview will be carefully explained in the PIS and the consent form. Participants will be able to pause the interview at any time, skip any questions and stop the interview entirely, or ask the interviewer to resume another day (this will be arranged wherever possible). Information will be made available at the end of the interview, and after more sensitive questions, for participants who recognise that they are feeling distressed. This will include resources signposting people to support services and helplines. The same risk protocol as above will be followed.

There might be some indirect benefits to the participant when taking part. People often value the opportunity to discuss their experiences and feelings. Further, people may feel keen to contribute to research concerning such a stressful and unprecedented situation. Various members of the research team have been running studies into mental health and well-being for many years,

including many tens of thousands of participants, and distress resulting from answering questions like the ones proposed in the current study is extremely rare.

## DISSEMINATION

We aim to rapidly disseminate summary findings to the senior management of participating Trusts and collaborating organisations such as NHS England in order to inform staff health and well-being strategies. Findings will be more broadly disseminated within the Trusts through their communication channels including websites and staff newsletters. Research findings will also be disseminated to NHS Trusts nationally via our professional network and professional bodies. In addition, findings will be published in academic journals, at conferences and stakeholder meetings and summaries placed on the dedicated study website.

**Correction notice** Author name, Sam Gnanapragasam, has been corrected. Colin Drummond has been added in Contributors section. Acknowledgements section and additional competing interest for Prof Wessely has been added.

**Acknowledgements** We wish to acknowledge the National Institute of Health Research (NIHR) Applied Research Collaboration East Midlands, and Professor Richard Morriss as part of the NIHR ARC National Collaboration NHS and Social Care Workforce Group, with NIHR ARCs East of England, South West Peninsula, South London, West, North West Coast, Yorkshire and Humber, and North East and North Cumbria. They enabled the set up of the national network of participating hospital sites and aided the research team to recruit effectively in the pandemic during such restrictions.

**Contributors** SW, NG, MH, RoR, RezR and SS are chief investigators of the study, and have contributed to manuscript drafts and approved the final draft. DL is a co-investigator of the study and has led the write-up of the manuscript. EC, IB and RG wrote the statistical analysis sections of the manuscript and provided comments on the draft manuscript. RB, HS, SH and ES are, respectively, the study project manager, postdoctoral researcher and research assistants, and have contributed to the manuscript drafts. AMR, SG and SM are co-investigators and have provided comments on the draft manuscript. DW is the study data manager and has provided comments on the draft manuscript. RebR is the TIDES study link and has provided comments on the draft manuscript. The NHS CHECK consortium includes the following co-investigators and collaborators: Sarah Dorrington, Ira Madan, Isabel McMullen, Martin Parsons, Catherine Polling, Danai Serfioti, Chloe Simela, Alexandra Pollitt, Rosie Duncan, Stephani Hatch, Daniel Leightley, Cerisse Gunasinghe, Paul Moran, Peter Aitken, Anthony David, Charlotte Wilson Jones, and Dominic Murphy. The NHS CHECK consortium includes the following site leads: Peter Jones, Jeremy Turner, Jesus Perez, Charles Goss, Richard Morriss, Adam Gordon, Frances Farnworth, Julian Walker, Mark Pietroni, Amy Dewar, Colin Drummond, Sean Cross, Mary Docherty, Scott Weich, David Levy, Ian Smith, Nusrat Husain, Robert Eames, Jessica Harvey, Chris Dickens, Damien Longson, Tayyeb Tahir, Peter Trigwell.

**Funding** Funding for NHS CHECK has been received from the following sources: Medical Research Council (MR/V034405/1); UCL/Wellcome (ISSF3/H17RCO/C3); Rosetrees (M952); Economic and Social Research Council (ES/V009931/1); as well as seed funding from National Institute for Health Research Maudsley Biomedical Research Centre, King's College London, National Institute for Health Research Health Protection Research Unit in Emergency Preparedness and Response at King's College London. This paper is independent research supported by the National Institute for Health Research ARC North Thames. The views expressed in this publication are those of the author(s) and not necessarily those of the National Institute for Health Research or the Department of Health and Social Care.

**Competing interests** MH, RoR and SW are senior NIHR Investigators. This paper represents independent research part-funded by the NIHR Maudsley Biomedical Research Centre Trust and King's College London (MH, SW, SS). SW has received speaker fees from Swiss Re for two webinars on the epidemiological impact of COVID 19 pandemic on mental health. The views expressed are those of the authors

and not necessarily those of the NHS, the NIHR, or the Department of Health and Social Care. RoR reports grants from DHSC/UKRI/ESRC COVID-19 Rapid Response Call, grants from Rosetrees Trust, grants from King's Together rapid response call, grants from University College London (UCL) (Wellcome Trust) rapid response call, during the conduct of the study; and grants from NIHR outside the submitted work. MH reports grants from DHSC/UKRI/ESRC COVID-19 Rapid Response Call, grants from Rosetrees Trust, grants from King's Together rapid response call, grants from UCL Partners rapid response call, during the conduct of the study; grants from Innovative Medicines Initiative and EFPIA, RADAR-CNS consortium, grants from MRC, grants from NIHR, outside the submitted work. SS reports grants from UKRI/ ESRC/DHSC, grants from UCL, grants from UKRI/MRC/DHSC, grants from Rosetrees Trust, grants from King's Together Fund, during the conduct of the study. NG reports a potential COI with NHSEI, during the conduct of the study; and is the managing director of March on Stress Ltd which has provided training for a number of NHS organisations, although it is not clear if the company has delivered training to any of the participating trusts or not as NG is not directly involved in commissioning specific pieces of work.

**Patient and public involvement**  Patients and/or the public were involved in the design, or conduct, or reporting, or dissemination plans of this research. Refer to the Methods section for further details.

**Patient consent for publication**  Not required.

**Provenance and peer review**  Not commissioned; externally peer reviewed.

**Open access**  This is an open access article distributed in accordance with the Creative Commons Attribution 4.0 Unported (CC BY 4.0) license, which permits others to copy, redistribute, remix, transform and build upon this work for any purpose, provided the original work is properly cited, a link to the licence is given, and indication of whether changes were made. See: https://creativecommons.org/licenses/by/4.0/.

**ORCID iDs**
Danielle Lamb http://orcid.org/0000-0003-1526-9793
Ewan Carr http://orcid.org/0000-0002-1146-4922
Simon Wessely http://orcid.org/0000-0002-6743-9929

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
