## [Reviewer comments · BMJ Open]

ARTICLE DETAILS

TITLE (PROVISIONAL)	NHS CHECK: protocol for a cohort study investigating the psychosocial impact of the COVID-19 pandemic on healthcare workers
AUTHORS	Lamb, Danielle; Greenberg, N; Hotopf, Matthew; Raine, Rosalind; Razavi, Reza; Bhundia, Rupa; Scott, Hannah; Carr, Ewan; Gafoor, Rafael; Bakolis, Ioannis; Hegarty, Siobhan; Souliou, Emilia; Rafferty, Anne Marie; Rhead, Rebecca; Weston, Danny; Gngangapragasam, Sam; Marlow, Sally; Wessely, Simon; Stevelink, Sharon

VERSION 1 – REVIEW

REVIEWER	Zimeras, Stelios Panepistemio Aigaiou, Statistics
REVIEW RETURNED	18-Apr-2021

GENERAL COMMENTS	In the paper an analysis of the protocol for COVID-19 for healthcare workers is presented. Some parts are unclear like: 1. Why the survey is taking place for 6 and 12 months.2. Sampling based on which methodology?3. For the questionnaires under which scale (for example Likert) would be constructed and under which protocol (reference)4. For the cohort analysis which statistical techniques would be proposed or applied for the analysis of the collected data. Finally it would be effective for somebody to understand the analysis a graphical presentation for every process.
---

REVIEWER	Kambouri, Katerina Democritus University of Thrace, Pediatric Surgery
REVIEW RETURNED	05-May-2021

GENERAL COMMENTS	it is an interesting and well designed protocol. it will be interesting for someone to see your results. i hope all the participants to cooperate
---

REVIEWER	Rolim Neto, Modesto Leite Federal University of Cariri
REVIEW RETURNED	02-Jun-2021

GENERAL COMMENTS	The title is accurate or relevant The aims of the study are clearly stated
--

	The study is original The study is useful and relevant to the aims of the Journal The design of the study is appropriate The sample size, selection and composition are appropriate Methods used to collect data (e.g. validated questionnaires and instruments, observational techniques) are appropriate Qualitative or quantitative methods used to analyse the data are appropriate Details of the methods (including settings and locations, procedures, dates of recruitment and follow-up or main outcomes) are clearly reported The data are less than 5 years old The study was approved by a research ethics committee prior to data collection Participants were asked for informed consent prior to data collection or informed consent was not required The qualitative or quantitative analyses were applied appropriately Missing data, e.g. non-respondents, drop-outs or non-responses, have been accounted for The results are clearly presented and explained No further qualitative or quantitative analysis is required The authors reflect on the strengths and limitations of the study The results are compared to related findings in the literature The results are discussed in relation to the relevant research, practice or policy issues The discussion and conclusions do not speculate beyond what has been shown in this study The article has a logical construction in a suitable format The article has an appropriate length (not unnecessarily long or too short to be useful) The writing is in a good standard of English, grammatically correct and easy to understand The abstract is in an unstructured format and is sufficiently informative Any tables and figures are all necessary, clearly annotated and easy to follow
--	--

VERSION 1 – AUTHOR RESPONSE

With regards to Reviewer 1's comments, I have:

1. Added clarification in the paper that the use of follow up surveys in the cohort study is in order to track outcomes over time.
2. Given more detail about sampling.
3. Clarified that all validated measures use Likert scale response options.

Regarding point 4, I believe this information has already been provided in the manuscript, under the 'Analysis' section. If there is further detail required please do let me know.